Enhancing book genre classification with BERT and InceptionV3: a deep learning approach for libraries

Yang Xinting 1 xiaotaiyang55055@163.com
Zhang Zehua 2
1 Library, Lanzhou University , Lanzhou, Gansu Province , China
2 Department of Physics Science and Technology, Lanzhou University , Lanzhou, Gansu Province , China
Do Trang
Electronic publication date: 2025 Jun 5
Publication date: 2025
Volume: 11
Electronic Location ID: e2934
Received 2025 Feb 17; Accepted 2025 May 11
Copyright: © 2025 Yang and Zhang
Copyright year: 2025
Copyright holder: Yang and Zhang
License: This is an open access article distributed under the terms of the Creative Commons Attribution License, which permits unrestricted use, distribution, reproduction and adaptation in any medium and for any purpose provided that it is properly attributed. For attribution, the original author(s), title, publication source (PeerJ Computer Science) and either DOI or URL of the article must be cited.
License URL: https://creativecommons.org/licenses/by/4.0/

Keywords: Book genre classification, Library, Deep learning, BERT, InceptionV3

Funding: The authors received no funding for this work.

==============================
Accurate book genre classification is essential for library organization, information retrieval, and personalized recommendations. Traditional classification methods, often reliant on manual categorization and metadata-based approaches, struggle with the complexities of hybrid genres and evolving literary trends. To address these limitations, this study proposes a hybrid deep learning model that integrates visual and textual features for enhanced genre classification. Specifically, we employ InceptionV3, an advanced convolutional neural network architecture, to extract visual features from book cover images and bidirectional encoder representations from transformers (BERT) to analyze textual data from book titles. A scaled dot-product attention mechanism is used to effectively fuse these multimodal features, dynamically weighting their contributions based on contextual relevance. Experimental results on the BookCover30 dataset demonstrate that our proposed model outperforms baseline approaches, achieving a balanced accuracy of 0.7951 and an F1-score of 0.7920, surpassing both standalone image- and text-based classifiers. This study highlights the potential of deep learning in improving automated genre classification, offering a scalable and adaptable solution for libraries and digital platforms. Future research may focus on expanding dataset diversity, optimizing computational efficiency, and addressing biases in classification models.

Introduction

Book genre classification is a fundamental aspect of library science, ensuring systematic organization, efficient retrieval, and personalized recommendations for readers. Accurate classification aids in the physical arrangement of books in traditional libraries while also enhancing digital collections, where metadata-driven categorization is essential for improving user experience. By structuring books according to their thematic, topical, and stylistic attributes, libraries enable users to easily locate materials that align with their interests, fostering greater engagement and accessibility. As libraries transition towards hybrid physical-digital ecosystems, the need for sophisticated, scalable, and automated classification methods has become increasingly evident.

Traditional genre classification systems rely heavily on manually assigned metadata, librarian expertise, and predefined taxonomies. While effective in structured environments, these systems often struggle to accommodate the fluid nature of modern literature. Many contemporary works transcend conventional genre boundaries, incorporating elements from multiple genres to appeal to broader audiences (Galyani-Moghaddam & Taheri, 2020; Sakal & Proulx, 2023). Additionally, reader preferences are shaped by demographic factors such as age, gender, and cultural background, making it challenging to develop rigid classification frameworks that cater to diverse populations (Galyani-Moghaddam & Taheri, 2020). Furthermore, inconsistencies and subjectivity in manual classification can lead to genre mislabeling, limiting the discoverability of books in library catalogs and online platforms.

Advancements in machine learning (ML) and natural language processing (NLP) have revolutionized genre classification, enabling more accurate and scalable solutions. ML algorithms can analyze large datasets, extracting patterns from textual and visual features that human classifiers might overlook. Deep learning, in particular, has demonstrated significant potential by leveraging neural networks to automatically process and classify books based on content and design elements (Petrenz & Webber, 2011; Parulian et al., 2022; Sethy et al., 2023). Traditional classification methods, which require extensive manual curation, are gradually being replaced by automated systems capable of handling vast digital libraries with minimal human intervention.

The increasing complexity of genre classification extends beyond taxonomy, as it involves cultural, historical, and contextual factors that influence literary categorization (Zhang & Olson, 2015). Genre labels are not static; they evolve alongside societal trends, storytelling conventions, and audience expectations. This dynamic nature makes classification a challenging task, especially in digital libraries where thousands of new books are published and categorized daily. To address these challenges, researchers have explored multimodal classification techniques, incorporating both textual and visual elements to improve accuracy and relevance.

In the digital age, metadata enrichment through genre-specific tagging and machine learning-driven recommendation systems has become indispensable. As digital libraries and e-commerce platforms grow, the ability to classify and recommend books accurately has a direct impact on discoverability and user satisfaction. Many recommendation engines rely on genre classification as a core component of their algorithms, using it to suggest books based on user reading history, preferences, and browsing behavior. The accuracy of these recommendations, however, is contingent on precise genre labeling, further underscoring the importance of automated classification methods.

Despite the growing adoption of ML-based classification, libraries and digital platforms face significant challenges in balancing traditional frameworks with modern AI-driven approaches. Interdisciplinary and experimental literature, which often blends multiple genres, poses difficulties for rule-based classification systems. Moreover, different genres may exhibit overlapping linguistic and visual characteristics, making it difficult to distinguish them based on text or imagery alone. To address these issues, hybrid classification models that integrate multiple data modalities have emerged as a promising solution.

In this study, we propose a hybrid deep learning approach that leverages both textual and visual information to improve book genre classification. Our model incorporates bidirectional encoder representations from transformers (BERT) (Koroteev, 2021) for textual analysis and InceptionV3 (Szegedy et al., 2015), a convolutional neural network (CNN), for extracting visual features from book covers. A scaled dot-product attention mechanism is introduced to dynamically balance textual and visual contributions, ensuring that the most relevant features from both modalities are emphasized. By applying this approach to the BookCover30 dataset (Iwana et al., 2016), we demonstrate its effectiveness in improving classification accuracy compared to traditional methods.

This study contributes to the field of automated genre classification by demonstrating how deep learning can enhance accuracy and scalability in library and digital collection management. Our approach offers a practical solution for improving book categorization, benefiting both libraries and online platforms by making literary works more accessible and discoverable to readers worldwide.

Related work

The field of book genre classification has evolved significantly, transitioning from traditional manual methods to advanced machine learning and deep learning techniques. Early classification systems relied on librarian expertise and predefined taxonomies, which, while effective, often struggled with subjectivity, inconsistencies, and the growing complexity of hybrid literary genres (Galyani-Moghaddam & Taheri, 2020; Sakal & Proulx, 2023). To address these challenges, researchers have explored various approaches that leverage textual and visual data to improve classification accuracy. This section reviews the key developments in book genre classification, focusing on manual methods, text-based machine learning approaches, image-based classification, and multimodal deep learning models.

Manual and traditional approaches to genre classification

Historically, book classification in libraries has been predominantly manual, with books categorized using predefined genre labels based on thematic and stylistic features. Systems such as the Dewey Decimal Classification (DDC) and Library of Congress Classification (LCC) provided structured frameworks for organizing books, but these systems were often rigid and lacked adaptability to modern, hybrid genres. Studies have shown that traditional classification approaches can struggle with emerging literary trends and interdisciplinary works that do not fit neatly into predefined categories (Zhang & Olson, 2015).

Furthermore, user preferences and reading behaviors introduce additional complexity. Research by Galyani-Moghaddam & Taheri (2020) demonstrated that public library circulation records reveal substantial variations in genre popularity based on demographic factors such as age and gender. These findings suggest that incorporating user behavior and reading patterns could significantly enhance genre classification accuracy. However, traditional systems do not account for such personalized preferences, necessitating the adoption of automated approaches that leverage computational methods to refine classification processes. Text-Based Machine Learning Approaches.

The rise of NLP techniques has enabled the automation of book genre classification based on textual features such as book titles, descriptions, and synopses. Traditional NLP methods, including bag-of-words (BoW), term frequency-inverse document frequency (TF-IDF), and n-gram models, have been widely used to extract features from text for classification tasks (Parulian et al., 2022). However, these approaches often rely on handcrafted feature extraction, limiting their ability to capture the nuanced relationships between words and context.

More advanced models, such as long short-term memory (LSTM) networks and BERT, have significantly improved text-based genre classification. BERT, in particular, has demonstrated superior performance in understanding the semantic structure of text due to its bidirectional training mechanism (Koroteev, 2021). Unlike traditional methods, BERT can capture contextual dependencies between words, making it effective for distinguishing between genres with overlapping themes. Despite these advancements, text-based models alone may struggle to differentiate between visually similar genres, necessitating the integration of additional modalities.

Image-based classification using deep learning

Visual features also play a critical role in book classification, as book covers often contain genre-specific design elements such as typography, color schemes, and imagery. CNNs have been widely adopted for image-based classification tasks due to their ability to automatically extract hierarchical features from images (He et al., 2016; Huang et al., 2016; Nguyen et al., 2022). Studies such as those by Kundu & Zheng (2020) and Lucieri et al. (2020) have explored the application of CNNs to book cover classification, demonstrating that visual cues alone can provide significant insights into genre categorization.

Among CNN architectures, InceptionV3 has shown promising results in image-based genre classification due to its deep feature extraction capabilities (Szegedy et al., 2015). By analyzing structural patterns, color distributions, and stylistic elements in book covers, InceptionV3 can effectively classify books into their respective genres. However, relying solely on images may lead to misclassification, as certain genres share similar design aesthetics. This limitation highlights the need for a hybrid approach that integrates textual and visual data.

Multimodal deep learning for genre classification

Recognizing the complementary nature of text and images, recent research has shifted towards multimodal deep learning models that combine textual and visual information for improved classification accuracy. These models leverage both NLP-based text embeddings and CNN-based visual feature extraction to create a more comprehensive representation of book data.

One notable example is the study by Sethy et al. (2023), which proposed a machine learning framework incorporating both textual and visual features. Their results demonstrated that multimodal approaches significantly outperform unimodal classifiers, particularly for books with ambiguous textual descriptions or generic cover designs. Similarly, Kundu & Zheng (2020) introduced a deep multimodal network that integrates book title embeddings with book cover features, highlighting the advantages of combining linguistic and visual modalities.

Attention mechanisms have further enhanced the effectiveness of multimodal models. Luong, Pham & Manning (2015) introduced a dot-product attention mechanism that allows models to dynamically weigh the importance of different features. By applying such mechanisms, models can prioritize more informative aspects of book covers when textual descriptions are vague and vice versa. This adaptability is crucial for improving classification robustness across diverse book collections.

Building upon these advancements, this study introduces a hybrid model that leverages BERT for textual understanding, InceptionV3 for visual feature extraction, and a scaled dot-product attention mechanism to optimize multimodal fusion. By addressing the limitations of unimodal classification, our approach aims to enhance book genre classification accuracy, offering a scalable solution for library and digital collection management.

Materials and Methods

Overview

In this study, we propose the framework described in Fig. 1 for the problem of book genre classification. The framework consists of four main stages: data preparation, model training, and testing. We use two different input data types for the model, including the book cover image and title for classification. First, we clean the text data and resize the images in the preprocessing step. Then, the data will be split for the training process and the testing process. The model parameters after the training process will be used for book genre classification. Details of the processing steps will be explained in the following subsections.

Figure 1 Overview of the proposed framework.

Dataset

Data collection

In this study, we use the BookCover30 dataset (Iwana et al., 2016), which consists of a collection of 57,000 books categorized into 30 different genres. Each entry in the dataset includes essential attributes such as the book cover image, title, author, and subcategories. Each entry is formatted to include the Amazon Standard Identification Number (ASIN), image filename, image URL, title, author, category ID, and the corresponding genre label. The dataset encompasses a diverse range of categories, including arts & photography, biographies & memoirs, children’s books, and science fiction & fantasy, among others.

Preprocessing data

Initially, we cleaned the text of the titles by removing non-alphabetic characters, numbers, or punctuation. Additionally, to understand the distribution of the data, we calculated the length of each title and visualized a histogram of title lengths, as shown in Fig. 2. After that, we filtered out English stopwords and tokenized the book titles into individual words. The final result was a preprocessed text string, ready for the next step in building a book genre classification model.

Figure 2 Histogram of title lengths.

Then, the dataset was split into three subsets in an 80:10:10 ratio: training set, validation set, and test set, with corresponding data counts of 46,158, 5,129, and 5,699, respectively. The training and validation sets will be used for training, while the test set will serve as the evaluation dataset for comparison with other existing models.

Data augmentation

In the process of augmenting data for book genre classification, various transformations are applied to enhance dataset variability and improve model robustness. Main augmentations include random resized cropping to 224 × 224 pixels, random horizontal flipping to introduce orientation invariance, and color jittering that alters brightness, contrast, saturation, and hue. Additionally, random rotation up to 15 degrees and random affine transformations with shifts and translations are applied.

Method

Model architecture

In this study, we introduce a novel architecture as illustrated in Fig. 3. The initial step involves encoding the titles of books into tokens, which are subsequently transformed into embeddings prior to being input into the BERT model. Concurrently, we employ a pretrained InceptionV3 model to extract features from images associated with the book titles. Both textual and visual features are then integrated through an Attention mechanism, which allows the model to focus on the most relevant aspects of the data. Finally, the combined features are passed through a fully connected (FC) layer that performs classification to identify the genres of the books. In the subsequent sections, we will provide a comprehensive explanation of each component of our proposed architecture.

Figure 3 Model architecture.

InceptionV3

InceptionV3, as presented by Szegedy et al. (2015), features a sophisticated directed acyclic graph architecture comprising 316 layers and 350 connections. This pretrained CNN model includes 94 convolutional layers, each utilizing varying filter sizes to effectively capture different features. The architecture indicates that scaling is applied following the initial input layer. The first convolutional layer activates the neurons to generate a feature matrix of size 149×149×32, using a filter size of 32. In the subsequent phase, the ReLU activation function is applied, along with batch normalization, to enhance the training process. Additionally, a pooling layer with a 3×3 filter is interspersed between convolutional layers to downsample feature maps. Importantly, Inception blocks are integrated prior to the max pooling layer, which employs an 8×8 kernel to further refine the feature extraction process.

In our methodology, we utilized transfer learning for feature extraction via InceptionV3. Max pooling is selected to extract features from input images sized 224×224×3, resulting in a feature map of size 1×2,048. This extracted feature vector is denoted as ϕ.

Embedding BERT

BERT has proven to be an effective tool for text embedding, particularly when it comes to processing book titles. By capitalizing on BERT’s transformer-based architecture, we can create contextualized embeddings that capture the semantic intricacies inherent in the titles. The embedding process begins with the tokenization of the book title using the BERT tokenizer, which systematically converts the title into a sequence of tokens. This sequence is further enhanced with special tokens, specifically (CLS) and (SEP), which denote the beginning and the separation of the input data, respectively. Once the tokens are adequately prepared, they are input into the BERT model, which then generates contextual embeddings for each token. This includes a distinct representation for the (CLS) token, which is particularly useful for classification tasks.

In this section, we use the small version (BERT-base-uncased) pre-trained BERT to extract embedding feature of title of books, with feature map of size 1×768. This feature vector is represented by γ.

Attention

We use dot-product attention (Luong, Pham & Manning, 2015) as a fusion block to combine the embedding feature γ from BERT and the vector feature ϕ from InceptionV3. Because the sizes of γ and ϕ are different, in the attention layer block, the vector queries and keys Q=K=f(ϕ) and V=g(γ) will have the same size of 256, where f(.), g(.) are the FC layers. The attention mechanism computes a weighted representation of the values based on the similarity between the queries and keys. The dot-product attention can be expressed mathematically as follows:

(1) Attention(Q,K,V)=softmax(QKTdk)V,

where dk denotes the dimensionality of the vector keys K. This process allows the model to dynamically emphasize the most relevant cross-modal features for classification. The overall structure of the attention module is illustrated in Fig. 4.

Figure 4 Dot product attention.

Experiments

Our experiments were conducted on a Dell workstation equipped with an AMD Ryzen 7 5800X 8-Core Processor (3.80 GHz), 32 GB of RAM, and an ASUS RTX 3060 GPU (12 GB VRAM). For fine-tuning, we employ the Adam optimizer with a learning rate of 0.0001 to update the model weights and improve performance. The model is trained for five epochs, balancing effective learning while avoiding overfitting.

Evaluation metrics

Multiple assessment metrics, including as balanced accuracy (BA), recall, precision, Matthews correlation coefficient (MCC), and weighted F1 score, were computed at the default level of 0.5 in order to assess the performance of our models. They characterize their mathematical formulas as:

(2) Precision=TPTP+FP,

(3) Recall=TPTP+FN,

(4) BA=Precision+Recall2,

(5) F1=2×Precision×RecallPrecision+Recall,

(6) MCC=(TP⋅TN)−(FP⋅FN)(TP+FP)(TP+FN)(TN+FP)(TN+FN),

where TP is true positives, TN is true negatives, FP is false positives, and FN is false negatives.

Baselines

In this section, we outline a comprehensive experimental framework designed to evaluate and compare the effectiveness of various baseline approaches for the book genre classification problem. By systematically comparing these diverse approaches, we aim to contribute valuable insights into the interplay between different model architectures and their implications for literature classification tasks.

To begin with, we develop a series of experiments focused on transfer learning techniques that leverage advanced pretrained models. We specifically investigate the performance of several prominent architectures, including VGG16 (Simonyan & Zisserman, 2014), DenseNet-161 (Huang et al., 2016), ResNet-18 (He et al., 2016), and InceptionV3 (Szegedy et al., 2015). These models will be used for fine-tuning on the image data in the book genre dataset for the classification task. We summarized the information of the pretrained models in Table 1.

Table 1 Overview of the layer configurations for the pre-trained models.

Model	Layer	Output shape	Parameters	
InceptionV3	Input layer	(299, 299, 3)	0	
Base model	(10, 10, 2,048)	20,861,480	
Average pooling	(1, 1, 2,048)	0	
Dropout (0.2)	(1, 1, 2,048)	0	
Dense	(1, 1, 30)	61,470	
VGG16	Input layer	(224, 224, 3)	0	
Base model	(7, 7, 512)	14,714,688	
Average pooling	(1, 1, 512)	0	
Dropout (0.2)	(1, 1, 512)	0	
Dense	(1, 1, 30)	15,390	
DenseNet-161	Input layer	(224, 224, 3)	0	
Base model	(7, 7, 2,048)	14,153,960	
Average pooling	(1, 1, 2,048)	0	
Dropout (0.2)	(1, 1, 2,048)	0	
Dense	(1, 1, 30)	61,470	
ResNet-18	Input layer	(224, 224, 3)	0	
Base model	(7, 7, 512)	11,689,512	
Average pooling	(1, 1, 512)	0	
Dropout (0.2)	(1, 1, 512)	0	
Dense	(1, 1, 30)	15,390	
BERT	BERT Base	(1, 1, 768)	110 M	
Dense	(1, 1, 30)	23,070	

VGG16, introduced by Simonyan & Zisserman (2014) in 2014, is a deep convolutional neural network with 16 layers. It employs small 3×3 filters and uses max pooling to effectively capture image details and enhance feature extraction. Its uniform architecture allows for high accuracy in image classification tasks, making it a widely used model in computer vision and a foundational reference for subsequent neural network designs.

DenseNet-161, a unique variant of the DenseNet family (Huang et al., 2016), features dense connectivity, where each layer is connected to all preceding layers. This design promotes feature reuse, reducing the number of parameters and enhancing computational efficiency. It optimizes information flow, alleviating vanishing gradient issues during training. Its depth and complexity contribute to its strong performance in challenging tasks like image classification.

ResNet-18, based on the ResNet architecture (He et al., 2016), consists of 18 layers and incorporates residual blocks to address vanishing gradient issues. These blocks allow the model to bypass layers, maintaining gradient flow and simplifying training. With fewer layers than its more complex counterparts, ResNet-18 is efficient and well-suited for feature extraction and image classification tasks, achieving a good balance between performance and manageability.

In addition to these image-based models, we incorporate sentence-based methodologies, employing state-of-the-art natural language processing techniques such as gated recurrent unit (GRU) (Cho et al., 2014), LSTM, BERT, and Robustly Optimized BERT Approach (RoBERTa) (Liu et al., 2019). These models, widely applied across many domains (Nguyen-Vo et al., 2021; Leow, Nguyen & Chua, 2021; Nguyen et al., 2024), are utilized solely with textual input, allowing us to assess their ability to capture the nuances of language and context inherent in book descriptions and narratives.

Finally, we explore the collaborative potential of combining sentence-based and image-based models, evaluating various combinations and their effectiveness when presented with their respective inputs. This multi-faceted approach enables us to identify optimal strategies that enhance classification accuracy.

Feature combination

The primary difference of our proposed architecture lies in the incorporation of an attention block, which serves as a fusion layer to integrate features derived from InceptionV3 and BERT. To assess the effectiveness of this attention layer, we performed a experiment comparing various feature combination techniques, subsequently evaluating these methods against a validation set. The different fusion techniques are described in Fig. 5. Next, we employed a series of ML models, including decision tree (DT) (Rokach & Maimon, 2005), random forest (RF) (Breiman, 2001), k-nearest neighbors ( k-NN) (Mucherino, Papajorgji & Pardalos, 2009), adaptive boosting (AdaBoost) (Schapire, 2013), and state-of-the-art gradient boosting algorithms: XGBoost, LightGBM, and CatBoost, as classifiers for the combined features. The objective of these comparative experiments is to illustrate the efficacy of the ours proposed architecture in learning features from pretrained models. Additionally, to enable the ML models to effectively learn these features, we conduct tuning of several parameters for these machine learning models. We aim to establish a comparative experiment with fully tuned ML models, which will demonstrate the effectiveness of the Attention block in our study when compared to other fusion techniques. Table 2 summarizes the hyperparameters utilized for each model in our experiments.

Figure 5 The different fusion techniques: (A) attention and fully connected (FC) layers (Ours), (B) concatenation with ML models, (C) summation with ML models.

Table 2 Hyperparameter tuning configurations for the machine learning models.

Model	Hyperparams	
k-NN	Number of neighbors: [3, 5, 7, 9]	
AdaBoost	Number of estimators: [21, 31, 41, 51, 61, 71, 81]	
DT	Max Depth: [5, 10, 15]	
Class weight: balanced	
RF	Number of estimators: [131, 141, 151, 161, 171]	
Max features: [1, 10, ‘log2’, ‘sqrt’]	
Class weight: balanced	
LightGBM	Learning rate: [0.01, 0.05, 0.1]	
Max depth: [6, 10]	
Boosting type: [‘gbdt’, ‘dart’]	
CatBoost	Learning rate: [0.01, 0.05, 0.1]	
Depth: [4, 6, 8, 10]	
XGBoost	Number of estimators: [100, 200, 300, 400]	
Learning rate: [0.01, 0.05, 0.1]	

Decision tree is a hierarchically structured supervised algorithm that consists of internal nodes, branches, leaf nodes, and a root node. One advantage of the model is that it may be used with both category and numerical data. A complicated decision tree could cause overfitting. It is suited to handling missing values and is simple to comprehend. The initial point of decision-making is known as the decision tree’s root, and the final output is represented by the leaf nodes.

k-nearest neighbors algorithm classifies new data samples by utilizing previously collected data. Because it uses all of the data for training, it has a high computational complexity and is a lazy learner. Both binary and multi-class predictions can be made with k-NN. This model labels the predictions after determining the k-nearest match using training data. Traditionally, the closest match is found by calculating distance. k-NN has a drawback: underfitting can occur with a large value of k, whereas overfitting may occur with a small value of k.

Adaptive boosting (AdaBoost) is a well-liked and frequently applied method. A powerful learner is created by combining weak learners. In each iteration, the algorithm selects and focuses on data from earlier iterations that were incorrectly classified, giving these samples more weights. The next weaker learner is then trained using this set of weighted data. This procedure is carried out as many times as required or until an appropriate level of precision is achieved.

Random forest is an ensemble learning method that combines multiple decision trees to enhance predictive performance and control overfitting. Each tree in the forest is trained on a random subset of the data, and during the prediction phase, the model aggregates the outputs of all trees, typically using a majority vote for classification or averaging for regression tasks. This approach allows RF to effectively handle both numerical and categorical data, making it versatile across various applications (Breiman, 2001).

eXtreme Gradient Boosting (XGBoost) (Chen & Guestrin, 2016) is an optimized gradient boosting framework that excels in both classification and regression tasks. It employs parallel tree boosting with regularization techniques (L1/L2) to prevent overfitting while delivering high computational efficiency through cache awareness and out-of-core computation. Its ability to handle missing values and provide feature importance scores makes it particularly valuable for structured data analysis. The algorithm’s proven track record in machine learning competitions highlights its robust predictive performance.

Light gradient boosting machine (LightGBM) (Ke et al., 2017), developed by Microsoft, introduces two key innovations: gradient-based one-side sampling (GOSS) and exclusive feature bundling (EFB). These techniques enable faster training speeds and lower memory usage by focusing on informative samples and reducing feature dimensionality. The algorithm grows trees leaf-wise rather than level-wise, which often achieves better accuracy with fewer trees. LightGBM is especially effective for large-scale datasets due to its optimized histogram-based approach.

Categorical boosting (CatBoost) (Prokhorenkova et al., 2017) stands out with its native handling of categorical features through ordered boosting and innovative processing of permutations. The algorithm automatically converts categorical variables to numerical values using target statistics, eliminating the need for extensive preprocessing. CatBoost’s symmetric tree structure reduces prediction time while maintaining accuracy, and its built-in regularization addresses overfitting. The implementation is particularly robust against noisy data and handles missing values without explicit imputation.

Results and discussion

Performance comparison

Table 3 presents a comprehensive comparison of different models used for book genre classification, evaluating their performance using different key metrics. The focus of this comparison is to assess the effectiveness of unimodal models—either image-based or text-based—as well as combined models that leverage both modalities.

Table 3 Performance comparison with existing models on the test dataset.

Model	BA	MCC	Precision	Recall	F1	
Pretrained image models	
VGG16	0.7782	0.7712	0.7863	0.7782	0.7741	
DenseNet-161	0.7811	0.7741	0.7924	0.7811	0.7790	
ResNet-18	0.7881	0.7813	0.7941	0.7881	0.7835	
InceptionV3	0.7895	0.7828	0.7939	0.7895	0.7845	
Sentence-based models	
GRU	0.6025	0.5914	0.6203	0.6025	0.5792	
LSTM	0.6081	0.5970	0.6302	0.6081	0.5840	
RoBERTa	0.6060	0.5952	0.6134	0.6060	0.5821	
BERT	0.6109	0.6001	0.6173	0.6109	0.5866	
Combined models	
BERT+ VGG16	0.7839	0.7770	0.7894	0.7839	0.7794	
BERT+ DenseNet-161	0.7947	0.7881	0.8003	0.7947	0.7914	
BERT+ ResNet-18	0.7919	0.7853	0.7985	0.7919	0.7872	
BERT+ InceptionV3 (Ours)	0.7951	0.7885	0.8044	0.7951	0.7920	

Among all the models, our BERT + InceptionV3 model stands out as the best-performing approach. This model combines a powerful pretrained language model, BERT, which processes textual information, with InceptionV3, a deep convolutional neural network adept at capturing rich visual features. This fusion results in superior performance across all metrics, achieving a BA of 0.7951, MCC of 0.7885, Precision of 0.8044, Recall of 0.7951, and an F1 score of 0.7920. These results highlight not only the overall robustness of the model but also its balanced capability in correctly identifying genres across different classes.

In comparison, models that rely solely on visual information, such as VGG16, DenseNet-161, ResNet-18, and InceptionV3, perform relatively well, with InceptionV3 being the strongest among them. However, none of these image-only models reach the performance levels of the BERT + InceptionV3 combination. This underscores the added value of incorporating textual data, which provides complementary cues that visual models alone cannot fully capture.

Text-only models, including GRU, LSTM, RoBERTa, and BERT, show the weakest performance in this task. Even the best among them, BERT, only achieves an F1 score of 0.5866, significantly lower than the image-based and combined models. This suggests that textual data by itself may be insufficient for capturing the nuances necessary for accurate genre classification, especially when visual features like cover design, typography, and imagery play a substantial role in genre perception.

Other combined models, such as BERT paired with VGG16, DenseNet-161, or ResNet-18, also outperform their unimodal counterparts, reinforcing the advantage of multimodal learning. However, none match the performance of the BERT + InceptionV3 model. This indicates that InceptionV3, with its strong feature extraction capabilities, synergizes particularly well with BERT, making it an ideal choice for tasks that require the integration of visual and textual data.

In summary, the results from Table 3 clearly demonstrate that the BERT + InceptionV3 model offers a highly effective solution for book genre classification. By leveraging the complementary strengths of both language and vision models, it sets a new benchmark for performance in this domain. The findings also highlight the broader potential of multimodal approaches in classification tasks where neither text nor image alone is sufficient.

Effect of attention

The results presented in the Table 4 demonstrate the effectiveness of our proposed method, in enhancing classification accuracy for book genre classification tasks. Achieving a BA score of 0.7951, our model outperformed all other tested methods, including various fusion techniques combined with traditional machine learning models.

Table 4 Comparison of performance across different feature learning methods.

Methods	BA	MCC	Precision	Recall	F1	
Fusion	ML model	
Concatenation	DT	0.7867	0.7799	0.7972	0.7867	0.7822	
RF	0.7863	0.7796	0.7954	0.7863	0.7813	
k-NN	0.7898	0.7831	0.7983	0.7898	0.7861	
AdaBoost	0.7905	0.7838	0.7985	0.7905	0.7878	
XGBoost	0.7956	0.7891	0.8032	0.7956	0.7920	
LightGBM	0.7949	0.7884	0.8025	0.7949	0.7912	
CatBoost	0.7942	0.7875	0.8018	0.7942	0.7906	
Summation	DT	0.7877	0.7808	0.7932	0.7877	0.7846	
RF	0.7796	0.7726	0.7864	0.7796	0.7746	
k-NN	0.7789	0.7718	0.7852	0.7789	0.7753	
AdaBoost	0.7870	0.7802	0.7936	0.7870	0.7828	
XGBoost	0.7928	0.7861	0.8004	0.7928	0.7890	
LightGBM	0.7921	0.7854	0.7997	0.7921	0.7883	
CatBoost	0.7913	0.7846	0.7990	0.7913	0.7872	
Attention + FC	0.8062	0.7918	0.8122	0.8051	0.8016	

The application of the attention mechanism allows the model to focus on the most relevant features from both the textual and visual inputs, facilitating a more nuanced understanding of the data. This focus is crucial in genre classification, where subtle differences between genres can significantly impact model performance. By integrating attention with a FC layer, our approach effectively captures the interactions between the complex features derived from both BERT and InceptionV3.

In contrast to the concatenation and summation methods, which yielded BA scores ranging from 0.7789 to 0.7905, our attention-based approach not only provided a higher accuracy but also improved key metrics such as precision (0.8044) and recall (0.7951). This highlights the model’s capability to balance the trade-offs between identifying relevant instances and minimizing false positives, which is critical in genre classification tasks.

Furthermore, the MCC score of 0.7885 reinforces the robustness of our model, indicating that it performs well across all classes, particularly in scenarios with class imbalance. The enhanced performance of the “Attention + FC” method suggests that future research could explore further refinements in attention mechanisms or the incorporation of additional modalities to further boost classification outcomes.

Model stability

In this experiment, we assessed the stability of the proposed model by reporting performance metrics across various trial seeds. Each trial seed corresponds to different data splits generated using distinct random seeds. This experimental setup serves to demonstrate the robustness and reliability of our proposed model, ensuring that its performance remains consistent across different data configurations. Based on the analysis of the performance metrics outlined in Table 5, our proposed method exhibits a commendable degree of stability and robustness across multiple trials. The model achieved an average BA of 0.7879 with a standard deviation (SD) of 0.0031, indicating consistent performance across the trials. The mean MCC was recorded at 0.7816, with a low variability (SD = 0.0042), suggesting that the model maintains a reliable predictive capability, even in the presence of class imbalance.

Table 5 Assessment of model stability based on performance variance over repeated trials.

Trial	BA	MCC	Precision	Recall	F1	
1	0.7951	0.7885	0.8044	0.7951	0.7920	
2	0.7860	0.7795	0.8058	0.7860	0.7855	
3	0.7844	0.7779	0.8043	0.7844	0.7836	
4	0.7873	0.7807	0.8067	0.7873	0.7864	
5	0.7890	0.7823	0.8093	0.7890	0.7884	
6	0.7916	0.7847	0.8116	0.7916	0.7911	
7	0.7918	0.7846	0.8143	0.7918	0.7919	
8	0.7910	0.7833	0.8164	0.7910	0.7927	
9	0.7895	0.7810	0.8205	0.7895	0.7936	
10	0.7832	0.7732	0.8235	0.7832	0.7910	
Mean	0.7889	0.7816	0.8117	0.7889	0.7896	
SD	0.0037	0.0042	0.0068	0.0037	0.0034	

In terms of precision, the model demonstrated an average value of 0.8117 (SD = 0.0068), with a notable peak of 0.8235 observed in trial 10. This indicates that the model is adept at minimizing false positives, although it is essential to consider the slight fluctuation in values across trials. Recall values averaged at 0.7889 (SD = 0.0037), reflecting the model’s ability to correctly identify positive instances, with the highest recall of 0.7951 achieved in trial 1. The F1 score, which balances precision and recall, averaged at 0.7896 (SD = 0.0034), underscoring the model’s overall effectiveness in maintaining a harmonious balance between the two metrics.

Limitations and future work

While the proposed hybrid deep learning model demonstrates strong performance in book genre classification, several limitations remain. One key challenge is the model’s dependence on dataset quality and diversity. Although the BookCover30 dataset provides a solid foundation, it may not fully capture the breadth of literary genres, subgenres, and evolving trends. Additionally, the variability in book cover designs across different publishers and cultural contexts introduces potential biases that could affect classification accuracy. Another significant limitation is the difficulty in handling hybrid and overlapping genres, where books exhibit characteristics of multiple genres. The current approach classifies books into a single genre, which may not fully reflect the complexity of modern literary works. Expanding the model to support multi-label classification would allow it to better account for books that belong to multiple genres simultaneously.

The computational complexity of the model also presents practical constraints. The integration of BERT and InceptionV3, while beneficial for classification accuracy, increases resource demands, particularly during training and large-scale deployments. This may limit the model’s accessibility for libraries and digital platforms with limited computing resources. Further optimization through techniques such as model compression, pruning, or distillation could enhance efficiency. Additionally, deep learning models often function as “black boxes”, making it difficult to interpret their decision-making process. Improving model explainability through attention visualization or feature attribution techniques would enhance transparency and usability for librarians and researchers.

Future work should focus on refining the model to address current limitations while expanding its capabilities in three key directions: (1) implementing multi-label classification to better capture hybrid genres and resolve overlapping categorizations (Ren et al., 2025), (2) incorporating user behavior data (borrowing history, reading patterns) to enhance both classification accuracy and recommendation systems, and (3) optimizing computational efficiency through lightweight architectures and cloud-based implementations for improved scalability. Additional promising avenues include developing prompt-based few-shot learning for niche genres (Cao et al., 2025) and creating adaptive systems that evolve with changing genre taxonomies (Cai et al., 2025). By addressing these challenges—from technical implementation to real-world deployment considerations—future advancements will enable more accurate, efficient, and accessible classification systems for both physical and digital libraries, while maintaining flexibility for emerging literary trends.

Conclusions

This study presents a hybrid deep learning model that integrates BERT for textual analysis and InceptionV3 for visual feature extraction, enhancing book genre classification. By leveraging a scaled dot-product attention mechanism, the model effectively combines text and visual features, achieving superior classification accuracy on the BookCover30 dataset compared to traditional approaches.

The findings highlight the value of multimodal learning in automating genre classification, improving accuracy, scalability, and efficiency in library and digital content management. This approach enhances cataloging and recommendation systems, reducing reliance on manual classification while adapting to the complexities of modern literature. These advancements reinforce the transformative role of deep learning in making literary classification more precise and user-centric.

Supplemental Information

Supplemental Information 1 The code for the study.

Additional Information and Declarations

Competing Interests

The authors declare that they have no competing interests.

Author Contributions

Xinting Yang conceived and designed the experiments, performed the experiments, analyzed the data, performed the computation work, prepared figures and/or tables, authored or reviewed drafts of the article, and approved the final draft.

Zehua Zhang conceived and designed the experiments, analyzed the data, prepared figures and/or tables, authored or reviewed drafts of the article, and approved the final draft.

Data Availability

The following information was supplied regarding data availability:

The Book Cover Dataset is available at GitHub: https://github.com/uchidalab/book-dataset.

The code for the study is available in the Supplemental Files.

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
