# Peer review of "Enhancing book genre classification with BERT and InceptionV3: a deep learning approach for libraries"

_PeerJ Computer Science, doi:10.7717/peerj-cs.2934_

## Round 0.1 · original submission · Major Revisions

· Academic Editor

Major Revisions

Please carefully address the reviewers' comments and revise the manuscript accordingly. In particular, Reviewer 2 has recommended incorporating additional models, which should be given special attention.

Reviewer 1 ·

Basic reporting

- The manuscript is written in clear and professional English, following formal expression and standards.
- The introduction and related works provide sufficient background and context for the study, with relevant literature appropriately cited and referenced.
- The manuscript's structure, formatting, and organization are clear.

Experimental design

- The proposed method is explained in sufficient detail to allow replication, including access to code, datasets, computing infrastructure, and reproduction scripts where applicable.
- Sources are adequately cited, and all references are properly quoted or paraphrased.
- The discussion on data preprocessing, evaluation methods, assessment metrics, and model selection techniques is clear and sufficiently detailed. However, the author may want to visualize the class distribution in the dataset in order to provide better insights into the dataset.

Validity of the findings

- The authors have conducted repeated experiments to demonstrate the model stability of the proposed method.
- When discussing “Limitations and Future Work”, the authors may want to discuss the computational complexity of the proposed model to offer a broader perspective on future research.

Additional comments

- In Table 1, the authors reference the model Xception, but it does not appear in subsequent results tables. Instead, the model InceptionV3 is mentioned. Could you clarify which model name is accurate for this study?

Reviewer 2 ·

Basic reporting

The authors propose a hybrid model that leverages BERT and InceptionV3 for book genre classification. The manuscript is well-structured, with informative sections, and the language is clear. The tables and figures are well-organized, providing detailed insights into the architecture, dataset, and results. However, as outlined below, several issues need to be addressed.

Experimental design

+ The proposed method is described comprehensively and clearly. The dataset and code are also provided. Key elements of the paper, such as data processing, evaluation metrics, and model selection, are well-implemented and thoroughly described.
+ The manuscript provides sufficient details regarding the computing infrastructure. However, information regarding the number of training epochs, optimizer function, and learning rate during model fine-tuning should be described in more detail.

Validity of the findings

+ Several machine learning models have been implemented to compare performance with the Attention module in the “Feature Combination” section. However, the authors should consider incorporating more powerful models, such as some gradient boosting algorithms like LightGBM, CatBoost, or XGBoost, to provide a more competitive comparison.
+ The comparison of pretrained image models is relatively comprehensive, covering architectures like VGG16, DenseNet-161, ResNet18, and InceptionV3. However, for sentence-based models, the authors should consider including additional baseline models such as GRU and/or RoBERTa. These models are well-suited for sentence-based tasks and have demonstrated competitive performance compared to LSTM and BERT. Expanding this comparison would offer a more comprehensive evaluation of the proposed method.
+ In addition, while the authors conduct experiments comparing model performance, they could also visualize feature representations of InceptionV3, BERT, and the combined InceptionV3+BERT model to highlight the differences and effectiveness of the proposed approach.

Additional comments

+ It seems that model “Xception” in Table 1 should be InceptionV3.

---

## Round 0.2 · accepted · Accept

· Academic Editor

Accept

The authors have thoroughly addressed all reviewer comments. Based on the reviewers' recommendations and my own evaluation, I confirm that the manuscript is ready for publication.

Reviewer 1 ·

Basic reporting

The paper is clear. No further comments.

Experimental design

All of my concerns have been satisfactorily addressed, and I have no further comments.

Validity of the findings

All looks good, nothing else to add.

Additional comments

No further changes are necessary.

Reviewer 2 ·

Basic reporting

The revised version is well-organized and clearly written. I don't have any other comments.

Experimental design

I have no further comments.

Validity of the findings

I have no further comments.

Additional comments

No further comments.